Reinvestigating the phylogeny of Myriapoda with more extensive taxon sampling and novel genetic perspective

Wang Jiajia 1
Bai Yu 1
Zhao Haifeng 2
Mu Ruinan 3
Dong Yan dongyan_bio@126.com 1
1 College of Biology and Food Engineering, Chuzhou University , Chuzhou , Anhui , China
2 Key Laboratory of Space Utilization, Technology and Engineering Center for Space Utilization, Chinese Academy of Sciences , Beijing , China
3 University of Chinese Academy of Sciences , Beijing , China
Gillespie Joseph
Electronic publication date: 2021 Dec 23
Publication date: 2021
Volume: 9
Electronic Location ID: e12691
Received 2021 Sep 2; Accepted 2021 Dec 5
Copyright: ©2021 Wang et al.
Copyright year: 2021
Copyright holder: Wang et al.
License: This is an open access article distributed under the terms of the Creative Commons Attribution License, which permits unrestricted use, distribution, reproduction and adaptation in any medium and for any purpose provided that it is properly attributed. For attribution, the original author(s), title, publication source (PeerJ) and either DOI or URL of the article must be cited.
License URL: https://creativecommons.org/licenses/by/4.0/

Keywords: Myriapoda, Phylogenetic relationships, Transcriptomic, Positive selection, Visual genes

Funding: The Natural Science Foundation of the Higher Education Institutions of Anhui Province KJ2018ZD041 The Key Program in the Youth Elite Support Plan in Universities of Anhui Province gxyqZD2020045 This work was supported by the Natural Science Foundation of the Higher Education Institutions of Anhui Province (KJ2018ZD041) and the Key Program in the Youth Elite Support Plan in Universities of Anhui Province (gxyqZD2020045). The funders had no role in study design, data collection and analysis, decision to publish, or preparation of the manuscript.

==============================
Background

There have been extensive debates on the interrelationships among the four major classes of Myriapoda—Chilopoda, Symphyla, Diplopoda, and Pauropoda. The core controversy is the position of Pauropoda; that is, whether it should be grouped with Symphyla or Diplopoda as a sister group. Two recent phylogenomic studies separately investigated transcriptomic data from 14 and 29 Myriapoda species covering all four groups along with outgroups, and proposed two different topologies of phylogenetic relationships.

Methods

Building on these studies, we extended the taxon sampling by investigating 39 myriapods and integrating the previously available data with three new transcriptomic datasets generated in this study. Our analyses present the phylogenetic relationships among the four major classes of Myriapoda with a more abundant taxon sampling and provide a new perspective to investigate the above-mentioned question, where visual genes’ identification were conducted. We compared the appearance pattern of genes, grouping them according to their classes and the visual pathways involved. Positive selection was detected for all identified visual genes between every pair of 39 myriapods, and 14 genes showed positive selection among 27 pairs.

Results

From the results of phylogenomic analyses, we propose that Symphyla is a sister group of Pauropoda. This stance has also received strong support from tree inference and topology tests.

Introduction

Myriapoda is a diverse group of terrestrial arthropods with more than 16,000 extant species (Moore, 2006) including millipedes and centipedes, which are familiar with our daily life. The presence of numerous legs (range from six to 750), which has given the myriapods their name, is obviously a symplesiomorphy (Marek & Bond, 2006). Myriapoda are widely distributed on all continents except Antarctica, and their diversity is concentrated in tropical and temperate regions, where you can find evidence of their habitat in soil, tree barks and trunks, fields and pastures, deserts, caverns, and coastal areas (Santos-Silva et al., 2019). There is an extensive debate on the sister group to monophyletic Myriapoda. Pancrustacea and Chelicerata are the two candidates with most support (Giribet & Edgecombe, 2019). There are four major classes of Myriapoda: Chilopoda (also known as centipedes, CHI), Diplopoda (also known as millipedes, DIP), Pauropoda (PAU), and Symphyla (SYM). Although the described extant species of all four classes are abundant, especially in CHI and DIP, the phylogenomic data are scarce (Szucsich et al., 2020). To date, only two phylogenomic studies have collected genome-wide or transcriptome-wide sequencing data covering all four classes for phylogeny investigation. Results from both studies supported monophyletic Myriapoda and the monophyly of each major class (DIP, CHI, PAU, and SYM) (Fernández, Edgecombe & Giribet, 2018; Szucsich et al., 2020; Bäcker, Fanenbruck & Wägele, 2008). However, the interrelationships among the four classes are still controversial. Previous molecular analyses proposed the PAU+SYM grouping (named Edafopoda), which strongly contradicted the sister-group relationship DIP+PAU (named Dignatha). And the hypothesis Edafopoda was supported by morphology and development (Regier et al., 2010; Regier & Zwick, 2011; Dong et al., 2012; Zwick, Regier & Zwickl, 2012; Miyazawa et al., 2014; Szucsich et al., 2020).

With the emergence of Next-Generation Sequencing (NGS) technology, some clarity has been gained in recent years. Fernández, Edgecombe & Giribet (2018) sequenced 12 myriapods, which greatly enriched the available data for phylogenomic analyses. Their results strongly support Dignatha topology with grouping PAU+DIP. A strong dependence on the choice of outgroups was emphasised in their study. Szucsich et al. (2020) generated 22 Myriapod RNA-Sequencing data by analysing 59 species. In addition to tree inference and outgroup selection impact testing, they conducted two topology tests: approximate unbiased (AU) tests and four-cluster likelihood-mapping (FcLM). Their results were consistent with Edafopoda topology, thereby grouping PAU+SYM (Szucsich et al., 2020). It is worth noting that both studies, although suggesting diverging topologies of the interrelationships among Myriapoda, placed Myriapoda as a sister group to Pancrustacea.

These two seminal works on the interrelationships of Myriapoda have laid a good foundation in phylogenomic data for further study. In our study, 60 species were investigated, including 39 Myriapoda members and 21 outgroups. We integrated data from the two aforementioned studies with three newly sequenced transcriptome data (one chilopod: Scolopendra sp.; and two diplopods: Epanerchodus sp., Skleroprotopus sp.). We compiled two concatenated supermatrices covering all four major classes of Myriapoda and three clades of outgroups, one including 20 gene partitions and the other, 369. We performed phylogenetic tree inference using Maximum-Likelihood method. The resulting trees had the same topology as Edafopoda (PAU+SYM) and a sister group of DIP+CHI, which was consistent with previous research results proposed by Szucsich et al. (2020). As shown in the previous study proposed by Szucsich et al., Pancrustacea is the closest relative to Myriapoda. Furthermore, topology tests including an AU test, weighted Kishino-Hasegawa (KH) test, and weighted Shimodaira-Hasegawa (SH) test were conducted on six topology hypotheses derived from the two most controversial phylogenetic relationships (Edafopoda and Dignatha). The results showed that almost all hypotheses derived from Dignatha were rejected with high probability. The topologies of PAU+SYM and DIP+CHI, which were determined from our best Maximum-Likelihood (ML) tree, survived all the tests. We attempted to find additional evidence to support PAU+SYM, and found that almost all species of PAU and SYM were small-sized, blind and soil-dwelling, which may have a significant impact on visual capabilities. For the evolution of vision-related genes, we performed Light Interaction Toolkit (LIT) gene identification on each of the 39 Myriapoda species and conducted positive selection analyses on the identified LIT genes. The distribution of LIT gene identification shared a very similar pattern among the four major classes, however, positive selection evidence was narrowed in CHI&DIP, CHI&PAU, CHI&SYM, DIP&PAU, and DIP&SYM.

Material and methods

Taxon sampling

Building upon previous works by Fernández, Edgecombe & Giribet (2018) and Szucsich et al. (2020), where more than 36 species representing the four major groups of myriapods were included in taxon sampling, we sequenced three additional species (one chilopod: Scolopendra sp.; and two diplopods: Epanerchodus sp., Skleroprotopus sp.) in this study. Our sampling was designed to maximise the representation of myriapod groups. Information on sampling localities and accession numbers in the Sequence Read Archive (SRA) database for each transcriptome is shown in Table 1, including four genomes from http://metazoa.ensembl.org. Twenty-one outgroups were also included: eight chelicerates (Liphistius malayanus, Centruroides vittatus, Damon diadema, Archegozetes longisetosus, Araneus diadematus, Egaenus convexus, Euscorpius sicanus, and Nymphon gracile), two onychophorans (Peripatopsis capensis and Peripatoides novaezealandiae), and 11 pancrustaceans (Daphnia pulex, Folsomia candida, Drosophila melanogaster, Eubranchipus grubii, Triops cancriformis, Nebalia bipes, Anaspides tasmaniae, Hemidiaptomus amblyodon, Tisbe furcata, Vargula hilgendorfii, and Xibalbanus tulumensis).

Table 1 Taxon sampling.

Species included in this study SRA accession numbers, information collection, and data sources are indicated.

Taxonomy	Clade alias	Species	DataType	Source	SRA #	Species alias		
Myriapoda, Chilopoda	CHI	Eupolybothrus cavernicolus	Transcriptome	Fernández et al. (2014)	ERR338470	Spe01	Eupolybothrus cavernicolus	
Myriapoda, Chilopoda	CHI	Cryptops hortensis	Transcriptome	Fernández, Edgecombe & Giribet (2016)	SRR1153457	Spe02	Cryptops hortensis	
Myriapoda, Chilopoda	CHI	Scutigera coleoptrata	Transcriptome	Fernández et al. (2014)	SRR1158078	Spe03	Scutigera coleoptrata	
Myriapoda, Chilopoda	CHI	Craterostigmus crabilli	Transcriptome	Fernández et al. (2014)	SRR3232915	Spe04	Craterostigmus crabilli	
Myriapoda, Chilopoda	CHI	Strigamia maritima	Genome	Chipman et al. (2014)	–	Spe05	Strigamia maritima	
Myriapoda, Chilopoda	CHI	Scolopendra sp.	Transcriptome	This study		Spe06	Scoropendra sp.	
Myriapoda, Chilopoda	CHI	Craterostigmus tasmanianus	Transcriptome	Szucsich et al. (2020)	SRR2774008	Spe31	Craterostigmus tasmanianus	
Myriapoda, Chilopoda	CHI	Henia illyrica	Transcriptome	Szucsich et al. (2020)	SRR3485986	Spe32	Henia illyrica	
Myriapoda, Chilopoda	CHI	Clinopodes flavidus	Transcriptome	Szucsich et al. (2020)	SRR1653181	Spe33	Clinopodes flavidus	
Myriapoda, Chilopoda	CHI	Himantarium gabrielis	Transcriptome	Szucsich et al. (2020)	SRR1653198	Spe34	Himantarium gabrielis	
Myriapoda, Chilopoda	CHI	Strigamia acuminata	Transcriptome	Szucsich et al. (2020)	SRR3485997	Spe35	Strigamia acuminata	
Myriapoda, Chilopoda	CHI	Schendyla carniolensis	Transcriptome	Szucsich et al. (2020)	SRR3485996	Spe36	Schendyla carniolensis	
Myriapoda, Chilopoda	CHI	Eupolybothrus fasciatus	Transcriptome	Szucsich et al. (2020)	SRR3485981	Spe37	Eupolybothrus fasciatus	
Myriapoda, Chilopoda	CHI	Eupolybothrus tridentinus	Transcriptome	Szucsich et al. (2020)	SRR3485982	Spe38	Eupolybothrus tridentinus	
Myriapoda, Chilopoda	CHI	Cryptops anomalans	Transcriptome	Szucsich et al. (2020)	SRR3485978	Spe39	Cryptops anomalans	
Myriapoda, Chilopoda	CHI	Scolopendra cingulata	Transcriptome	Szucsich et al. (2020)	SRR1653235	Spe40	Scolopendra cingulata	
Myriapoda, Chilopoda	CHI	Scolopocryptops rubiginosus	Transcriptome	Szucsich et al. (2020)	SRR1653236	Spe41	Scolopocryptops rubiginosus	
Myriapoda, Diplopoda	DIP	Glomeris marginata	Transcriptome	Fernández, Edgecombe & Giribet (2016)	SRR3233211	Spe11	Glomeris marginata	
Myriapoda, Diplopoda	DIP	Narceus americanus	Transcriptome	Fernández, Edgecombe & Giribet (2016)	SRR3233222	Spe12	Narceus americanus	
Myriapoda, Diplopoda	DIP	Eudigraphis taiwanensis	Transcriptome	Fernández, Edgecombe & Giribet (2016)	SRR3458640	Spe13	Eudigraphis taiwanensis	
Myriapoda, Diplopoda	DIP	Cyliosoma sp.	Transcriptome	Fernández, Edgecombe & Giribet (2016)	SRR3458641	Spe14	Cyliosoma sp.	
Myriapoda, Diplopoda	DIP	Brachycybe sp.	Transcriptome	Brewer & Bond (2013)	SRR945430	Spe15	Brachycybe sp.	
Myriapoda, Diplopoda	DIP	Epanerchodus sp.	Transcriptome	This study		Spe16	Epanerchodus sp.	
Myriapoda, Diplopoda	DIP	Skleroprotopus sp.	Transcriptome	This study	SRR1145732	Spe17	Skleroprotopus sp.	
Myriapoda, Diplopoda	DIP	Callipus foetidissimus	Transcriptome	Szucsich et al. (2020)	SRR3485975	Spe42	Callipus foetidissimus	
Myriapoda, Diplopoda	DIP	Craspedosoma sp. [AD-2016]	Transcriptome	Szucsich et al. (2020)	SRR3485977	Spe43	Craspedosoma sp. [AD-2016]	
Myriapoda, Diplopoda	DIP	Haploglomeris multistriata	Transcriptome	Szucsich et al. (2020)	SRR3485985	Spe44	Haploglomeris multistriata	
Myriapoda, Diplopoda	DIP	Glomeridella minima	Transcriptome	Szucsich et al. (2020)	SRR3485983	Spe45	Glomeridella minima	
Myriapoda, Diplopoda	DIP	Ommatoiulus sabulosus	Transcriptome	Szucsich et al. (2020)	SRR1653222	Spe46	Ommatoiulus sabulosus	
Myriapoda, Diplopoda	DIP	Thalassisobates littoralis	Transcriptome	Szucsich et al. (2020)	SRR1653242	Spe47	Thalassisobates littoralis	
Myriapoda, Diplopoda	DIP	Polydesmus complanatus	Transcriptome	Szucsich et al. (2020)	SRR3485993	Spe48	Polydesmus complanatus	
Myriapoda, Diplopoda	DIP	Polyxenus lagurus	Transcriptome	Szucsich et al. (2020)	SRR3485994	Spe49	Polyxenus lagurus	
Myriapoda, Diplopoda	DIP	Polyzonium germanicum	Transcriptome	Szucsich et al. (2020)	SRR3485995	Spe50	Polyzonium germanicum	
Myriapoda, Pauropoda	PAU	Pauropus huxleyi	Transcriptome	Fernández, Edgecombe & Giribet (2018)	SRR6145369	Spe10	Pauropus huxleyi	
Myriapoda, Pauropoda	PAU	Acopauropus ornatus	Transcriptome	Szucsich et al. (2020)	SRR3485973	Spe51	Acopauropus ornatus	
Myriapoda, Symphyla	SYM	Scutigerella sp	Transcriptome	Fernández et al. (2014)	SRR3458649	Spe07	Scutigerella sp	
Myriapoda, Symphyla	SYM	Hanseniella sp.	Transcriptome	Fernández, Edgecombe & Giribet (2016)	SRR6217953	Spe08	Hanseniella sp.	
Myriapoda, Symphyla	SYM	Symphylella sp.	Transcriptome	Fernández, Edgecombe & Giribet (2018)	SRR6144316	Spe09	Symphylella sp.	
Myriapoda, Symphyla	SYM	Hanseniella nivea	Transcriptome	Szucsich et al. (2020)	SRR3485984	Spe52	Hanseniella nivea	
Chelicerata	CHE	Liphistius malayanus	Transcriptome	Sharma et al. (2014)	SRR1145736	Spe18	Liphistius malayanus	
Chelicerata	CHE	Centruroides vittatus	Transcriptome	Sharma et al. (2014)	SRR1146578	Spe19	Centruroides vittatus	
Chelicerata	CHE	Damon diadema	Transcriptome	Szucsich et al. (2020)	SRR3485979	Spe25	Damon diadema	
Chelicerata	CHE	Archegozetes longisetosus	Transcriptome	Szucsich et al. (2020)	SRR1653174	Spe26	Archegozetes longisetosus	
Chelicerata	CHE	Araneus diadematus	Transcriptome	Szucsich et al. (2020)	SRR3485974	Spe27	Araneus diadematus	
Chelicerata	CHE	Egaenus convexus	Transcriptome	Szucsich et al. (2020)	SRR3485980	Spe28	Egaenus convexus	
Chelicerata	CHE	Euscorpius sicanus	Transcriptome	Szucsich et al. (2020)	SRR1653192	Spe29	Euscorpius sicanus	
Chelicerata	CHE	Nymphon gracile	Transcriptome	Szucsich et al. (2020)	SRR1653221	Spe30	Nymphon gracile	
Onychophora	ONY	Peripatopsis capensis	Transcriptome	Szucsich et al. (2020)	SRR1145776	Spe23	Peripatopsis capensis	
Onychophora	ONY	Peripatoides novaezealandiae	Transcriptome	Szucsich et al. (2020)	SRR3485992	Spe24	Peripatoides novaezealandiae	
Crustacea	PAN	Daphnia pulex	Genome		–	Spe20	Daphnia pulex	
Crustacea	PAN	Folsomia candida	Genome		–	Spe21	Folsomia candida	
Crustacea	PAN	Drosophila melanogaster	Genome		–	Spe22	Drosophila melanogaster	
Crustacea	PAN	Eubranchipus grubii	Transcriptome	Szucsich et al. (2020)	SRR1653190	Spe53	Eubranchipus grubii	
Crustacea	PAN	Triops cancriformis	Transcriptome	Szucsich et al. (2020)	SRR1653248	Spe54	Triops cancriformis	
Crustacea	PAN	Nebalia bipes	Transcriptome	Szucsich et al. (2020)	SRR1653215	Spe55	Nebalia bipes	
Crustacea	PAN	Anaspides tasmaniae	Transcriptome	Szucsich et al. (2020)	SRR1653173	Spe56	Anaspides tasmaniae	
Crustacea	PAN	Hemidiaptomus amblyodon	Transcriptome	Szucsich et al. (2020)	SRR1653196	Spe57	Hemidiaptomus amblyodon	
Crustacea	PAN	Tisbe furcata	Transcriptome	Szucsich et al. (2020)	SRR1653244	Spe58	Tisbe furcata	
Crustacea	PAN	Vargula hilgendorfii	Transcriptome	Szucsich et al. (2020)	SRR1811940	Spe59	Vargula hilgendorfii	
Crustacea	PAN	Xibalbanus tulumensis	Transcriptome	Szucsich et al. (2020)	SRR1653240	Spe60	Xibalbanus tulumensis	

RNA extraction and sequencing

Following the manufacturer’s instructions, total RNA was extracted using a commercial RNA extraction kit (TAKARA). Samples were treated with Ambion turbo DNA-free DNase to remove residual genomic and rRNA contaminants during mRNA purification. The quantity and quality (purity and integrity) of mRNA were assessed using a NanoDrop ND-2000 UV spectrophotometer (Thermo Fisher Scientific).

For mRNA sequencing library preparation, mRNA was first enriched and purified with oligo (dT)-rich magnetic beads and then broken into short fragments, followed by paired-end sequencing on an Illumina Hiseq 4000 platform.

Data processing and de novo assembly

Sequencing adaptors and low-quality sequences were trimmed using Trimmomatic v 0.36 (Bolger, Lohse & Usadel, 2014) with default parameters. The clean data were assembled with Trinity (release 2.11.0) with 100 GB memory and a path reinforcement distance of 50 (Grabherr et al., 2011). The redundancy of all assembled transcripts was removed using CD-HIT v. 4.8.1 (under the cd-hit-est mode, with default parameters Li & Godzik, 2006). TransDecoder v5.5.0 (https://github.com/TransDecoder/TransDecoder) was then utilised for nucleotide sequence translation and longest ORF selection.

Orthology assignment and phylogenetic matrix construction

Both classical pipeline (OrthoFinder) and single-copy gene selection method (Benchmarking Universal Single-Copy Orthologs; BUSCO) were used in orthology assignment among the 60 selected taxa. Orthogroups were first identified using OrthoFinder v 2.2.7 (Emms & Kelly, 2015) with default settings (BLASTP E value ≤ 1e−5 and MCL inflation parameter of 1.5). Single-copy genes in arthropods were identified in our datasets with BUSCO v4.1.4 with default settings (E value ≤ 1e−6) based on hidden Markov model profiles, where BUSCO dataset arthropoda_odb10 (https://busco-data.ezlab.org/v4/data/lineages) was used as reference (Seppey, Manni & Zdobnov, 2019). For each BUSCO and each taxon, the longest hit of duplicated BUSCO homologous genes was retained for further analysis.

Putative orthogroup filtering was based on the gene occupancy threshold, which means that an orthogroup (or BUSCO) was selected if it could be found in more than or equal to the threshold number of taxa. For example, a 50% gene occupancy threshold would select orthogroups that were present in ≥ 50% of the included taxa. We selected two thresholds of 100% and 90% gene occupancy to obtain information on most species and to minimise the computational burden. Protein sequences of each orthogroup were aligned with MAFFT v 7.305b (maxiterate was set to 1000 in globalpair mode) prior to concatenation (Katoh & Standley, 2013). Then, Aliscore v2.0 was used to perform each orthogroup’s multiple sequence alignment (MSA) for ambiguous or randomly aligned sections’ identification, followed by Alicut v2.2 for error section’s trimming (Kück, 2009). After filtering with 100% and 90% gene occupancy thresholds, two raw matrices were constructed using custom Python scripts. Finally, the matrices were trimmed using MARE (v 0.1.2) to select optimised data subsets from the supermatrices for phylogenetic inference (Meyer, Meusemann & Misof, 2011).

Best partition schemes finding

The best-fit partitioning schemes and models of evolution for phylogenetic analyses were searched and estimated with PartitionFinderV2.0.0 (Lanfear et al., 2017), where amino acid substitution models were restricted to LG, WAG, JTT, and BLOSUM62. The corrected Akaike Information Criterion (AICc) model was selected and was set under greedy search.

Phylogenetic tree inference

Maximum-likelihood tree

Tree searches were performed for the two supermatrices with a ML approach, using IQ-TREE (v1.6.12) with the above best partitioning schemes. Statistical support was derived from 100 non-parametric slow bootstrap replicates. Replicates to perform SH-like approximate likelihood ratio test were set to 1000 and unsuccessful iterations to stop were set to 300. The initial tree searches were set from a completely random tree (Nguyen et al., 2015). The full command was: ‘iqtree -s MSAmatrix.phy -alrt 1000 -b 100 -t RANDOM -spp partition_ file. nex-nstop 300.’

Bootstrap support inference

Bootstrapping analyses were applied with RAxML-NG v1.0.0 (Kozlov et al., 2019), and the autoMRE bootstrap convergence test was set for a sufficient number of replicates. The bootstrap support was then mapped onto the phylogenetic trees using RAxML v8.2.11 (Stamatakis, 2014).

Topology testing

To evaluate support for the different hypotheses concerning the relationship proposed by previous studies among the four major classes within Myriapoda, topology tests were run for each dataset using the corresponding best partition scheme with IQ-TREE (v1.6.12), where the AU-test, weighted KH-test, and weighted SH-test were included, and all tests performed 100,000 resamplings using the resampling of estimated log-likelihoods (RELL) method (Nguyen et al., 2015). Six proposed hypotheses representing the two most controversial phylogenetic relationships (Edafopoda and Dignatha) of four major classes in Myripoda were compared in our topology tests; detailed information is shown in Fig. 1.

Figure 1 Hypotheses on relationships of the major myriapod lineages Chilopoda, Diplopoda, Symphyla and Pauropoda.

Hypothesis Eda. 1, Hypothesis Eda.2 and Hypothesis Eda.3 are three quartet topologies derived from Edafopoda, which grouping the PAU and SYM as a sister clade; Hypothesis Dig.1, Hypothesis Dig.2 and Hypothesis Dig.3 are three quartet topologies derived from Dignatha, which grouping the PAU and DIP as a sister clade.

Hypothesis Eda.1(topology: ((CHI,DIP), (SYM, PAU));).

Hypothesis Eda.2(topology: (CHI, (DIP, (SYM, PAU)));).

Hypothesis Eda.3(topology: (DIP, (CHI, (SYM, PAU)));).

Hypothesis Dig.1(topology: ((CHI, SYM), (DIP, PAU));).

Hypothesis Dig.2(topology: (CHI, (SYM, (DIP, PAU)));).

Hypothesis Dig.3(topology: (SYM, (CHI, (DIP, PAU)));).

Identification of LIT genes

A modified version of the phylogenetically informed annotation tool pipeline named PIA2 (https://github.com/xibalbanus/PIA2) was applied for the identification of visual opsins in this study, and the parameters were set default (Pérez-Moreno et al., 2018). In this pipeline, 111 genes from the LIT, a collection of genes that underlie the function or development of light-interacting structures in metazoans, representing 13 different parts in visual pathways (photoreceptor specification, retinal determination network, phototransduction, rhabdomeric, phototransduction, ciliary, retinoid pathway, vertebrate, retinoid pathway, invertebrate, melanin synthesis, pterin synthesis, ommochrome synthesis, heme synthesis, crystallins, diurnal clock, and opsin), was taken as reference (Speiser et al., 2014). We applied the pipeline to protein sequences of each species.

Identification of positively selected genes

Evidence of positive selection was indicated by estimating the ratios of nonsynonymous substitutions (Ka or dN) and synonymous substitutions (Ks or dS), also called substitution rates (Ka/Ks or dN/dS value). The coding sequence of each identified LIT genes was aligned between a pair of taxa separately with MAFFT v 7.305b with default settings. And then the substitution rate was calculated using the KaKs_calculator with the following settings, method of calculation: GMYN, genetic code table: The Echinoderm and Flatworm Mitochondrial Code (Wang et al., 2010).

Results

Transcriptome assembly and phylogenomic dataset construction

NGS technologies have empowered phylogenomic analyses in the last few decades. It has dramatically increased the size of datasets applied to phylogenetic questions. Within the framework of combining NGS technologies and phylogenomic techniques, we decided to re-investigate Myriapoda phylogeny with three newly sequenced species. Combined with published data from two outstanding studies on Myriapoda phylogeny, the data from a total of 60 species (39 from Myriapoda, 8 Chelicerata, 11 Crustacea, and 2 Onychophora) were used in this study (Table 1) (Fernández, Edgecombe & Giribet, 2018; Szucsich et al., 2020). Except for the four species with published genome data, raw reads of the remaining 56 species were trimmed and assembled de novo. Orthology assignments of the 60 species were mainly based on BUSCO results, and more than 70% of the ortholog gene set (BUSCO dataset: arthropoda_odb10, comprising 1013 single-copy protein-coding genes or ortholog group, OG) were identified in 56 species (details in Table S1). Additionally, 786 of the BUSCO orthology assignments were confirmed using OrthoFinder (details in Table S1).

Concatenated supermatrices were compiled using a threshold of percentage gene occupancy of 100% and 90% (Fig. 2) (González et al., 2015). We found that 32 OGs were represented in all 60 species (100% gene occupancy), of which 20 were confirmed by OrthoFinder. There were 505 OGs represented in more than 54 species (90% gene occupancy), and 369 OG assignments were confirmed by OrthoFinder. Thus, two datasets comprising 20 and 369 OGs were obtained (Table S1). After MSA, identification, and removal of ambiguously aligned sections in each dataset, two phylogenomic supermatrices on the amino acid levels were constructed, which are hereafter referred to as OCC100 and OCC90. Matrix OCC100 included 20 gene partitions and spanned 8,401 aligned sites with 0.395 overall information content. Matrix OCC90 included 369 gene partitions and spanned 129,085 aligned sites with 0.318 overall information content.

Figure 2 Schematic of the two supermatrices used in this study.

Matrix OCC100 was based on the blue section (100% gene occupancy) where 32 BUSCOs were included, and matrix OCC90 was based on the purple and blue section (>90% gene occupancy) where 505 BUSCO were included.

Phylogenetic tree inference and topology analysis

We constructed Maximum-Likelihood (ML) trees based on the best partition schemes and best-fitting substitution models schemes with matrices OCC100 and OCC90. Three main results were found from the inferred trees. All the analyses recovered Myriapoda as the monophyletic sister group of Pancrustacea with high support (Fig. 3). As for the relationships among the four myriapod classes: Symphyla (SYM), Chilopoda (CHI), Diplopoda (DIP), and Pauropoda (PAU), we found a sister group relationship of CHI+DIP, and another sister group relationship of PAU+SYM (Fig. 3). Both were highly supported by the bootstrap result (PAU+SYM: 100%, DIP+CHI: 100%). The three newly sequenced species (Scolopendra sp., Epanerchodus sp., and Skleroprotopus sp.) were positioned in the expected clades.

Figure 3 Best ML tree on matrix OCC90.

Best Maximum-Likelihood tree inferred with IQ-TREE derived from matrix OCC90 (60 taxa, alignment length: 129,085 amino acid positions, 369 gene partitions), and rooted with Onychophora, where all the topology were consistent with the best ML tree inferred from matrix OCC100. Statistical support was derived from 1,000 non-parametric bootstrap replicates, trees were converged after 700 replicates.

A variety of groupings of the Myriapoda classes have been proposed, where two hypotheses, Edafopoda and Dignatha, received the most attention (Fig. 1). Edafopoda is a grouping of PAU+SYM which has been supported by shared genetic sequences (Fig. 1). However, in Dignatha, the PAUs were positioned with the DIPs. In this study, all trees inferred were congruent with the unrooted quartet topology with CHI+DIP and PAU+SYM (Hypothesis Eda.1, Fig. 1). We conducted three types of topology tests—AU test, KH test, and weighted SH test—on the quartet topology of Edafopoda and Dignatha, where four different phylogenomic datasets were applied. The results consistently supported the topology Hypothesis Eda.1, which is the only topology to not be rejected in any test (Table 2). Almost all hypotheses derived from Dignatha were rejected with high significance, especially in the phylogenomic matrix OCC90 (Table 2). When comparing the results from the two phylogenomic matrices, we found that all testing results of matrix OCC100 were consistent with and covered by that of matrix OCC90. Under matrix OCC90, the datasets that were different in outgroup selection (PAN or CHE) exclusively showed divergence when rejecting Hypothesis Eda.2, where all three topology tests on the datasets with CHE were not rejected, which was completely opposite to the results of datasets with PAN (Table 2). In other words, we found that the sister group of Edafopoda (PAU+SYM) received less support from CHI than the clade of DIP+CHI, but it cannot be completely denied.

Table 2 Results of topology tests.

Results of approximately unbiased (AU), weighted Kishino-Hasegawa (KH), and weighted Shimodaira-Hasegawa (SH) tests comparing historically proposed hypotheses of the inner relationships of Myriapoda. A total of 100,000 RELL replicates were performed for each test, plus signs (+) denote the 95% confidence sets (not rejected), minus signs (-) denote significant exclusion (rejected).

Occ100	Pau-test	Pkh-test	Psh-test	
	ExcludeCHE	ExcludePAN	ExcludeCHE	ExcludePAN	ExcludeCHE	ExcludePAN	
Hypothesis Eda.1	0.6200 +	0.2110 +	0.5300 +	0.1320 +	1.0000 +	0.5310 +	
Hypothesis Eda.2	0.2950 +	0.9320 +	0.2330 +	0.8680 +	0.6130 +	1.0000 +	
Hypothesis Eda.3	0.0101 -	0.1630 +	0.0341 -	0.1010 +	0.1490 +	0.4840 +	
Hypothesis Dig.1	0.0088 -	0.1210 +	0.0925 +	0.0832 +	0.1630 +	0.1440 +	
Hypothesis Dig.2	0.0805 +	0.0372 -	0.1380 +	0.0194 -	0.3380 +	0.1360 +	
Hypothesis Dig.3	0.5410 +	0.0302 -	0.4700 +	0.0177 -	0.8100 +	0.0398 -	
Occ90	Pau-test	Pkh-test	Psh-test	
	ExcludeCHE	ExcludePAN	ExcludeCHE	ExcludePAN	ExcludeCHE	ExcludePAN	
Hypothesis Eda.1	1.0000 +	0.7770 +	1.0000 +	0.7770 +	1.0000 +	1.0000 +	
Hypothesis Eda.2	3.62E−38 -	0.2230 +	0.0000 -	0.2230 +	1.00E−05 -	0.5570 +	
Hypothesis Eda.3	2.85E−50 -	4.15E−87 -	0.0000 -	0.0000 -	0.0000 -	0.0039 -	
Hypothesis Dig.1	9.72E−121 -	4.44E−110 -	0.0000 -	0.0000 -	0.0000 -	0.0000 -	
Hypothesis Dig.2	2.21E−91 -	5.15E−55 -	0.0000 -	0.0000 -	0.0000 -	0.0000 -	
Hypothesis Dig.3	1.09E−37 -	3.04E−11 -	0.0000 -	0.0000 -	1.00E−05 -	0.0000 -	

Outgroup dependence of myriapod phylogeny inference

Despite the quartet topology of CHI+DIP and PAU+SYM being recovered in our analyses, the relationships among the four major classes in Myriapoda varied across phylogenomic datasets, with dependence on outgroup selection proposed in previous studies (Fernández, Edgecombe & Giribet, 2018; Szucsich et al., 2020). Given that the phylogeny inference was sensitive to outgroup choice, we conducted topology tests on datasets with different clades of outgroups: one with only CHE as outgroup, and the other with only PAN (Figs. S2–S5). ML tree inference of the former resulted in a sister group relationship of CHI+DIP and another sister group relationship of PAU+SYM, which was congruent with the results inferred from the datasets with the full taxon sampling (Fig. 3). However, we found that the ML tree inferred from the latter datasets resulted in a sister group relationship of CHI and DIP, with SYM as a sister to this clade, followed by PAU. Although these quartet topology results were also obtained in previous studies, negligible support could be obtained from bootstrapping analyses (Szucsich et al., 2020).

LIT genes’ identification in Myriapoda

Using the PIA2 pipeline, we identified 2,001 transcripts in 39 Myriapoda species as putative components involved in the development of light-interacting structures, including 96 LIT genes, which are important components of 11 visual pathways (Fig. 4). A total of 13 visual pathways were compiled in the pipeline, and two were absent in this study (Table 3, retinal determination network and opsin synthesis) which involved 10 LIT genes. In addition, the other five absent LIT genes were Gq_gamma, RBP3, Dat, TYR, and reflectin_1a. We investigated the completeness of the amino acid level of each ortholog by calculating the ratio of the length of the identified peptide to the target reference peptide, as depicted in Fig. 4; the lighter the cell-filling colour, the more incomplete the transcript. As shown in Fig. 4, the distribution patterns of the identified LIT genes among the four classes are very similar; LIT genes from prc (photoreceptor specification), reti (retinoid pathway, invertebrate), heme (heme synthesis), and crys (crystallins) were rarely identified in Myriapoda (shown as a large blank area in Fig. 4). We also found that LIT genes from reti (retinoid pathway, invertebrate, 0), ommo (ommochrome synthesis, 2), and clock (diurnal clock, 2) were rarely identified in PAU (Table 3). As for the common LIT genes among these four classes, we found that 62 LIT genes could be identified in at least one of the four classes (Fig. 5C), and three (GC, TH, KF) could be identified in all 39 myriapods investigated. The following three genes were separately involved in three different pathways, ctrans: phototransduction in ciliary, mel: melanin synthesis, and ommo: ommochrome synthesis. We compared the LIT genes co-identified between DIP and CHI, and PAU and SYM according to the visual pathways in which the genes participated (Fig. 5B and Fig. 5A). We found that clock (diurnal clock), crys (crystallins), ommo (ommochrome synthesis), and reti (retinoid pathway, invertebrate) were more abundant in the sister group of DIP and CHI.

Figure 4 LIT genes identified in four major subgroups of Myriapoda.

Tree structure on the left of the figure was the best ML tree in this study. The colorful cells represent the completeness on amino-acid level of each ortholog by calculating the ratio of the length of the identified peptide and the target reference one provided in the PIA2. The lighter of the cell filling color, the more incomplete the transcript. Abbreviations of the visual pathways are following, PRC, Photoreceptor Specification; RTRANS, Phototransduction, Rhabdomeric; CTRANS, Phototransduction, Ciliary; RETV, Retinoid Pathway, Vertebrate; RETI, Retinoid Pathway, Invertebrate; MEL, Melanin Synthesis; PTER, Pterin Synthesis; OMMO, Ommochrome Synthesis; HEME, Heme Synthesis; CRYS, Crystallins; CLOCK, Diurnal Clock.

Table 3 Distribution of LIT genes identification.

Statistical results of LIT gene identification. Sum: the sum of transcripts from a specific class that was identified as LIT genes involved in a specific visual pathway. Max: the maximum quantity of transcripts from a species among a specific class that was identified as the LIT genes involved in a specific visual pathway. Mean: ratio of the sum and species quantity of a specific class.

	CHI	DIP	PAU	SYM	
	unique	sum	max	unique	sum	max	unique	sum	max	unique	sum	max	
rdn	0	0	0	0	0	0	0	0	0	0	0	0	
prc	8	20	8	10	35	8	6	6	5	10	12	9	
rtrans	12	179	12	12	170	12	12	24	12	12	44	12	
ctrans	11	139	11	12	153	11	10	19	10	10	31	10	
retv	7	102	7	7	99	7	7	13	7	7	24	7	
reti	4	8	4	5	18	4	0	0	0	5	10	5	
mel	8	116	8	8	100	8	7	13	7	8	23	8	
pter	8	81	8	8	56	8	6	12	6	8	18	8	
ommo	7	48	7	7	53	7	1	2	1	7	13	7	
heme	7	27	7	8	22	7	7	14	7	7	8	7	
crys	5	34	4	5	22	4	1	2	1	1	4	1	
clock	12	114	12	12	87	10	6	6	6	11	20	10	
opsin	0	0	0	0	0	0	0	0	0	0	0	0	
	CHI	DIP	PAU	SYM	
	unique	sum	max	mean	unique	sum	max	mean	unique	sum	max	mean	unique	sum	max	mean	
rdn	0	0	0	0.00	0	0	0	0.00	0	0	0	0.00	0	0	0	0.00	
prc	8	20	8	1.18	10	35	8	2.19	6	6	5	3.00	10	12	9	3.00	
rtrans	12	179	12	10.53	12	170	12	10.63	12	24	12	12.00	12	44	12	11.00	
ctrans	11	139	11	8.18	12	153	11	9.56	10	19	10	9.50	10	31	10	7.75	
retv	7	102	7	6.00	7	99	7	6.19	7	13	7	6.50	7	24	7	6.00	
reti	4	8	4	0.47	5	18	4	1.13	0	0	0	0.00	5	10	5	2.50	
mel	8	116	8	6.82	8	100	8	6.25	7	13	7	6.50	8	23	8	5.75	
pter	8	81	8	4.76	8	56	8	3.50	6	12	6	6.00	8	18	8	4.50	
ommo	7	48	7	2.82	7	53	7	3.31	1	2	1	1.00	7	13	7	3.25	
heme	7	27	7	1.59	8	22	7	1.38	7	14	7	7.00	7	8	7	2.00	
crys	5	34	4	2.00	5	22	4	1.38	1	2	1	1.00	1	4	1	1.00	
clock	12	114	12	6.71	12	87	10	5.44	6	6	6	3.00	11	20	10	5.00	
opsin	0	0	0	0.00	0	0	0	0.00	0	0	0	0.00	0	0	0	0.00	

Figure 5 Comparison of the LIT genes in four main subgroups of Myriapoda.

Co-identified LIT genes grouping by visual pathways between PAU and SYM (A), DIP and CHI (B). Number of distinct LIT genes identified among each monophyletic subgroup of Myriapoda.

Selection tests on LIT genes

To test whether genes associated with the evolution of light interactions in Myriapoda have undergone potentially adaptive changes, Ka/Ks calculations were conducted. A total of 23,832 aligned LIT gene pairs were calculated, including 8,717 pairs from CHI&DIP, 2,048 from CHI&SYM, 1,220 from CHI&PAU, 1,158 from DIP&PAU, 1,981 from DIP&SYM, 278 from PAU&SYM, 4,241 from CHI&CHI, 3,981 from DIP&DIP, 47 from PAU&PAU, and 161 from SYM&SYM. Positive selection was detected in 27 pairs, indicated by Ka/Ks ≥1, five pairs from CHI&DIP, one from CHI&PAU, three from CHI&SYM, two from DIP&PAU, two from DIP&SYM, seven from CHI&CHI, and seven from DIP&DIP. As depicted in Fig. 5, no evidence of positive selection for LIT genes was found in PAU&PAU, SYM&SYM, PAU&SYM. Values of Ka/Ks in the range of 0.5 to 1.0, which indicates relaxed selection, were observed in 395 pairs, covering all classes combinations (details in Table S2). The remaining 23,435 pairs had Ka/Ks values ranging from 0.0002 to 0.5, representing 98% of the pairs we calculated, which means that most of the genes in the four major classes were under purifying selection. Positive selection was detected in the following 14 LIT genes: clot, Cnga1, CSAD, DAGK, DDC, Galpha_it, GC, Gprk1, Gq_alpha, Pde6abc, PKC, PLC, RBP1, RDH8, timeless, and trp, which cover the clock, ctrans, mel, pter, retv, and rtrans visual pathways (Fig. 6). In addition, the transient receptor potential protein trp, which encodes a component of the rhabdomeric phototransduction pathway, was identified and positively selected in PAU&CHI, SYM&CHI, and SYM&DIP.

Figure 6 Positive selection support for LIT genes by KaKs calculation.

Subgroup distribution of the positively selected genes, the bigger the bubbles, the more pairs found under positive selection, and the smallest bubbles mean the support was one.

Discussion

In this study, we first performed phylogenomic analyses on Myriapoda with three newly sequenced members by integrating phylogenetic tree inference and topology testing. Our results showed that CHI+DIP and PAU+SYM were the best quartet topologies for interrelationships among the four major classes. This is consistent with the recent study by Szucsich et al. (2020) but in conflict with the earlier results published by Fernández, Edgecombe & Giribet (2018) and with morphological evidence. We provided an innovative point in taxon sampling, which was the addition of three newly sequenced Myriapoda species with high-quality sequencing, two of which were members of DIP and one of CHI. It is particularly worth mentioning that the taxon sampling of the PAU class in this study was represented by two species (Pauropus huxleyi and Acopauropus ornatus), instead of just one specie in previous studies which may increase the risk of mispositioning PAU in quartet topology. Previous studies have shown that, in molecular studies, the more extensive a taxon sample collection, the more convincing the phylogeny results (Fernández, Edgecombe & Giribet, 2018; Szucsich et al., 2020).

There seems to be no end to the debate about the interrelationships among the four main Myriapoda classes, though phylogenomic analyses do provide robust evidence for phylogeny, where novel research points were being summoned. We then turned our study’s focus on the common environmental and ecological habits and conditions of most species of PAU and SYM. The pauropods (PAU) inhabit a variety of soil types, but sometimes found in plant litter and decaying logs, and they cannot burrow themselves, which makes them be confined to crevices and tunnels already present (Adis et al., 2002). Besides, they are small-sized with little mobility, which can be the reason for that they rarely appear outside the Amazonian sampling sites (Battirola et al., 2018; Hilgert et al., 2019; Santos-Silva et al., 2019). Though symphylans (SYM) most often are true soil-dwellers, they can live in many different habitats: in leaf litter, in the upper humus layer, and in pure soil, both in upper layers and in the mineral subsoil (Adis et al., 2002). In summary, most species of PAU and SYM are small-sized, soil-dwelling and blind, while most of CHI and DIP are eyed. It cannot be denied that these characters are shared with some orders of DIP and CHI. The order Geophilomorpha (comprise over 1,254 species) and the family Cryptopidae (order Scolopendromorpha, comprise over 184 species) of the class CHI (comprise over 4,142 species) are both live underground and blind (Adis et al., 2002), which accounts for a third of the CHI. In class DIP, order Polydesmida, order Platydesmida, order Glomeridesmida separately comprise over 3,500, 50 and 30 species, which accounts for almost a half of the DIP (comprise over 7,753 species) (Adis et al., 2002). However, almost all species of PAU (comprise over 835 species) and SYM (comprise over 197 species) were blind and soil-dwelling (Adis et al., 2002). It could be due to biases in species richness, but we insisted that visual capability was a good innovation point, and made a preliminary exploration. With the help of analytical pipeline developed for whole-genome wide identification of visual genes (Speiser et al., 2014; Pérez-Moreno et al., 2018), we identified visual genes for each of the 39 Myriapoda species separately, and compared the distribution of positively selected genes among the four major classes. As our results showed, positive selection was detected between species from CHI and DIP, CHI and PAU, CHI and SYM, DIP and PAU, DIP and SYM, but none was found between PAU and SYM. Both, the LIT gene identification and the positive selection, indicated that the components of the rhabdomeric phototransduction pathway, which are employed by the photoreceptors found in the eyes of many invertebrates, received the most attention (Figs. 5 and 6). The majority of the components of the rhabdomeric phototransduction pathway are responsible for conferring light sensitivity to photoreceptors from the retinas of cephalopods (Mitchell & Mayeenuddin, 1998; Kishigami et al., 2001; Murakami & Kouyama, 2008), including Gq protein, r-opsins, and transient receptor potential protein. The positive selection pressure on trp in PAU&CHI, SYM&CHI, and SYM&DIP could reflect adaptive changes in the machinery during the rhabdomeric phototransduction pathway. In this respect, the positive selection signatures on the components of the rhabdomeric phototransduction pathway in Myriapoda could be regarded and further analysed from a broader perspective. Besides, blind species account for more than a third in both CHI and DIP, thus LIT genes’ patterns between eyed CHI and blind CHI, eyed DIP and blind DIP, warrants further study.

Although phylogenomic data covering 39 myriapods have been published, the acquisition of more sufficient data is still expected, especially transcriptomic data from species in PAU and SYM. To address the interrelationships of the four major classes of Myriapoda, we consolidated most of the available data for Myriapoda phylogeny analyses, and conducted a series of phylogenomic analyses, which provided strong evidence for the PAU+SYM topology of Edafopoda. To find other evidence for Edafopoda, we identified visual genes that detected positive selection pressure among the four major classes of Myriapoda. In short, our analyses offered more information to further promote the study of interspecific evolutionary relationships among Myriapoda.

Conclusion

For the highly disputed interrelationships of Myriapoda, our best phylogenetic tree involved 39 species favored the hypothesis Edafopoda, which was supported by a series of topology tests we conducted and consistent with plenty of previous studies. The commonness of living habits was investigated among the four major classes, and we made a preliminary exploration by LIT analyses. Though weak evidence was found to support the monophyly of PAU and SYM, we think it is a good research point which needs further study.

Supplemental Information

Supplemental Information 1 Ortholog group assignment

Click here for additional data file.

Supplemental Information 2 Ka/Ks results of LIT genes

Click here for additional data file.

Supplemental Information 3 ML tree on matrix OCC100

Click here for additional data file.

Supplemental Information 4 ML tree on matrix OCC90 without outgroups from Pancrustacea

Click here for additional data file.

Supplemental Information 5 ML tree on matrix OCC100 without outgroups from Pancrustacea

Click here for additional data file.

Supplemental Information 6 ML tree on matrix OCC90 without outgroups from Chelicerata

Click here for additional data file.

Supplemental Information 7 ML tree on matrix OCC100 without outgroups from Chelicerata

Click here for additional data file.

Additional Information and Declarations

Competing Interests

Author Contributions

Data Availability

The authors declare there are no competing interests.

Jiajia Wang analyzed the data, prepared figures and/or tables, authored or reviewed drafts of the paper, and approved the final draft.

Yu Bai analyzed the data, prepared figures and/or tables, and approved the final draft.

Haifeng Zhao performed the experiments, prepared figures and/or tables, and approved the final draft.

Ruinan Mu performed the experiments, prepared figures and/or tables, authored or reviewed drafts of the paper, and approved the final draft.

Yan Dong conceived and designed the experiments, authored or reviewed drafts of the paper, and approved the final draft.

The following information was supplied regarding data availability:

The raw reads data are available at the Short Read Archive (SRA) and BioProject: PRJNA758760.

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
