# Peer review of "Reinvestigating the phylogeny of Myriapoda with more extensive taxon sampling and novel genetic perspective"

_PeerJ, doi:10.7717/peerj.12691_

## Round 0.1 · original submission · Major Revisions

Dear Dr. Wang and colleagues:

Thanks for submitting your manuscript to PeerJ. I have now received three independent reviews of your work, and as you will see, one reviewer recommended rejection, while the others suggested minor revisions (but with plenty of suggested changes to be made). I am affording you the option of revising your manuscript according to all three reviews but understand that your resubmission may be sent to at least one new reviewer for a fresh assessment (unless the reviewer recommending rejection is willing to re-review).

The major problem raised by Reviewer 2 is in your treatment of visual capabilities and terrestrial environment. This seems addressable.

Please fix all of the identified issues with text, spelling, etc.

Therefore, I am recommending that you revise your manuscript, accordingly, taking into account all of the issues raised by the reviewers.

I look forward to seeing your revision, and thanks again for submitting your work to PeerJ.

Good luck with your revision,

-joe

·

Basic reporting

.

Experimental design

.

Validity of the findings

.

Additional comments

Review of submission #64880, by Wang et al., to PeerJ

The submission in review is a very nice, smart and useful paper, easy reading and finely illustrated, that sheds new light on the phylogenetic interrelations between four major classes of Myriapoda, using a rich array of candidate outgroups. I would immediately recommend publication had it not been for a few reproaches, all easy to consider though.

The flaws as I see them are mainly rooted in the authors’ apparent negligence in taxonomy. Why term the CLASSES Chilopoda, Diplopoda, Symphyla, and Pauropoda either as “subfamilies” (a distinct taxonomic category, albeit applied wrong) or “subgroups” (not wrong, but inexact)?
It would also seem advisable to show the sampling set (Table 1) in a clearer taxonomic context (e.g. structured as per order). Thus, there are 16 orders (taxonomic categories!) among extant Diplopoda, and five in Recent Chilopoda, but whereas your taxon listing includes representatives of all five orders of Chilopoda, that of Diplopoda contains members of only nine, i.e. far from complete and thus not too representative (especially so as regards blind taxa).

Indeed, among Diplopoda there are several ORIGINALLY and TOTALLY blind orders such as Polydesmida or Platydesmida, representatives of both of which are listed in Table 1. The same concerns the entire order Geophilomorpha or the family Cryptopidae (order Scolopendromorpha) of the class Chilopoda. What about their LIT genes versus those of the “eyed” orders? It has been very clever for the authors to consider the soil as a major environmental factor to have affected the evolution of the blind and mostly soil-dwelling Symphyla and Pauropoda which are reconfirmed as being particularly close phylogenetically, but, as noted above, several orders (the Polydesmida is the largest and most diverse amongs all Diplopoda!) or families (taxa!) of Myriapoda, both soil-dwelling and epigean, are likewise blind and thus requiring special attention (or at least mention if not a brief discussion) as regards the LIT data.

A single typo spotted:
Line 320, not Scoropendra but Scolopendra

Only two small lapses noted, as follows:
Line 438, italicize “Koenig” as well, since the Bonn Museum is named after Alexander Koenig
Line 440, it must read “Bonner Zentrum für Molekulare Biodiversitätsforschung (ZMFK)” (NB: Two Umlaut as diacritics)!

As a verdict, I believe the paper is outstanding and definitely merits publication, yet this being possible only upon MINOR revision. I do not think another round of peer reviewing is necessary.

I do not mind if my name as that of a reviewer is revealed to the authors.

Moscow, September 8th, 2021
Sergei Golovatch

Reviewer 2 ·

Basic reporting

This is a neat paper that reconstructed the phylogeny of Myriapoda based on transcriptomic data with high support values. However, some major weaknesses made me reject this manuscript.

1. Introduction of the background is insufficient. For example, the authors should explain the difference of habitat between Pauropoda + Symphyla and Chilopoda + Diplopoda in detail.

2. Citations need to be modified. For instance, the citations of the sentence in lines 44-46 were confused.

3. Some expressions need to be more specific. For example, in lines 33-34, the authors said ‘Myriapoda…including millipedes, centipedes, and others’. Myriapoda is the topic of this manuscript and there are only four classes in Myriapoda. Why not introduce them all?
In lines 34-35, the description of the diagnostic characteristic of Myriapoda is ambiguous. In accordance with this state, is a spider with 8 legs a myriapod?
In line 39, the authors said ‘the described extant species of all four subgroups are abundant’. It’s wrong. Actually, only Chilopoda and Diplopoda are species-rich, and Pauropoda and Symphyla are much less speciose (Szucsich et al. 2020).

3. Some terms were erroneously used. For instance, “Dignatha” in line 44 should be changed as “Edafopoda” and “Edafopoda” in line 45 should be changed as “Dignatha” (replace these two terms).
The authors regarded the four subgroups of Myriapoda as ‘subfamily’ (in lines 282, 287, 288, 331, 332, 334, 352, 362, 367). This is wrong. The taxonomic category of Diplopoda, Chilopoda, Symphyla and Pauropoda is class rather than subfamily.
In line 140, if the authors used Akaike Information Criterion then the abbreviation should be AIC. AICc stands for corrected Akaike Information Criterion.

4. Sentences in lines 358-359, ‘...instead of just one species in previous studies. This may increase the risk of mispositioning PAU in quartet topology’ is better to changed as ‘...instead of just one species in previous studies which may increase the risk of mispositioning PAU in quartet topology’

5. The word ‘and’ in line 45 was not typed in Times New Roman.

6. The first ‘CHI’ in Supplementary_Figure_1 should be ‘CHE’.

Experimental design

The fatal flaw of this manuscript is in this part. The authors state that the habitat of Pauropoda and Symphyla is soil, and it may impact their visual capabilities. However, actually, all four subgroups of Myriapoda are terrestrial arthropods and inhabiting the soil. Why the influence only occurs in Pauropoda and Symphyla, and disappears in Chilopoda and Diplopoda? Furthermore, even if there is a difference in visual ability between Pauropoda + Symphyla and Chilopoda + Diplopoda, it also could be a consequence of convergent evolution. Therefore, the visual ability couldn’t provide evidence for the monophyly of Pauropoda and Symphyla.

Validity of the findings

no comment.

Additional comments

no comment.

·

Basic reporting

The English language is generally clear, unambiguous and professional. However, I found the following ambiguous or imprecise points, which deserve corrections or clarifications:
- the 4 clades (Chilopoda, Symphyla, Diplopoda, and Pauropoda) are variously called subgroups or subfamilies in different paragraphs of the article; however, they are not "subfamilies" in the current and traditional taxomomy, accordig to established conventions of taxonomic nomenclature;
- lines 44-45: the names Dignatha and Edafopoda are exchanged erroneously;
- line 45: "which strongly contradicted the phylogenetic relationships (sister group of DIP+PAU, named Edafopoda)"; the meaning is unclear. Perhaps the meaning is: "which contradicted the sister-group relationship DIP+PAU (named Edafopoda)".
- "Scolopendra" is sometimes misspelled "Scoropendra"
- line 116: "We confirmed the orthology relationships of the 60 selected taxa". The wording is unclear.
- lines 178-183: brackets and other signs at the end of the lines are excessive
- line 199: "The substitution rate [...] was aligned": the wording should be more precise
- line 238: "in the right clade"; avoid "right/wrong", because the true phylogeny is unknown, and we have only hypotheses;
- Table 1: "Ailas" misspelled?
- Table 2: some abbreviations in the table (e.g. Pau-test) do not correspond to the abbreviations in the legend (AU)
- line 349: "phylogenomic analyses on Myriapoda phylogeny"; wording redundant
- line 388: "intraspecific" perhaps is misspled for "interspecific"

The literature references are sound and adequate. However, I invite the authors to consider the state of knowledge on the ecological and habitat diversity of Chilopoda and Diplopoda, if they want to keep presenting the LIT gene analysis as useful and informative on the phylogeny between the four clades [but see my comments below].

The general structure of the article is standard, but there are many redundant paragraphs that duplicate information. For instance:
- the methods and results are reported already in too much detail in the Introduction;
- the methods are repeated in too much detail in the Result section

The trees in suppl. fig. 2-5 appear without bootstrap values, but the authors wrote (line 269) "negligible support could be obtained from bootstrapping". Authors should show statistical supports for these trees.

Experimental design

The major merit and originality of this study is to have integrated two independent datasets, previously analysed independently, in a single analysis, so that to increase the taxon sampling.

Validity of the findings

The phylogenetic analysis is well done, sound and rigourous.
Also the LIT gene analysis appears well done, rigorous and very original for the Myriapoda. However, it is not fully clear how the LIT gene analysis can contribute to the general explicit aim of the study (clarify the phylogentic relations among the four clades).
The rationale of employing the LIT gene analysis to add support to the phylogenetic hypothesis should be explained more clearly and more precisely by the Authors.
If, instead, the LIT gene analysis does not bear on the phylogenetic reconstruction, it should be presented not as an additional tool for the assessment of phylogenetic hypotheses, but merely as a complementary, orginal evaluation of the evolution of vision and vision-related genes in the Myriapoda as a whole.
In the present manuscript, the following senteces should be provided with arguments or, alternatively, revised:
- line 23: "provide a new perspective to observe the above-mentioned question, since we also identified the visual genes..."
- line 28: "From [...] the new perspective gained by visual genes, we propose that Symphyla is a sister group of Pauropoda ..."
- line 73: "We attempted to find additional evidence to support PAU+SYM and found
that their common habitat is soil".
- line 369: "The respective grouping of PAU and SYM on this basis could make sense"


Moreover, authors do not present and not discuss the results of the Bayesian analyses. These results should be reported , at least briefly, and the trees provided in the Suppl. Mat. at least.

When describing the phylogenetic results (lines 235 and following), the author describe only the topology obtained from the OCC90 matrix, but they should tell explicitly also that the OCC100 matrix produced a different topology, contraddicting the Edafopoda hypothesis.

Additional comments

Some statements appear partly questionable, or imprecise, or subjective. So, I invite the authors to reconsider them and avoid any statement that sounds subjective or opinion non based on evidence.
In particular:
- line 34: "The most distinct myriapod characteristic is that their number of legs range from 6 to 750". Actually, it seems subjective. One can say that another character is "the most distinct", e.g., the elongate shape of the body, or the structure of the mouth parts, or the adaptation to terrestrial environment... Moreover, the range of variation of leg number is very huge, and so hardly "diagnostic" (=distinguishing Myriapopa from all other animals) and other animal groups share similar and partially overlapping numbers of legs (Onychopora, some crustacean groups, ...).
- line 74: ".. PAU+SYM and found that their common habitat is soil". This is also true for most Chilopoda and Diplopoda. Even if the author refer to strict endogeic life (inside the soil, without epigeic activity), they shoud consider that large subclades of Chilopoda and Diplopoda are structly endogeic and blind, like PAU + SYM.
- "Our sampling was designed to maximise the representation of all four myriapod groups" is erroneous, because all available genomic datasets have been used, without any selection, and the new sequenced taxa were not chosen to maximize the representation of the 4 clades (of the 3 taxa, 2 were of the same clade DIP).
- line 354: "The most innovative point of our study was the addition of three...". Actually, the addition of three new transcriptomic datasets is not so crucial for the study, because they are from three species belonging to the two largest clades (Chilopoda and Diplopoda), and in particular to species well nested within the clades (not basally diverging subclades), and not belonging to the focal clade of the study (Pauropoda).
- line 359: "We believe that in molecular studies, the more extensive a taxon sample collection, the
more convincing the phylogeny results". This is well established, by means of many studies. Authors should presnet this statement as an evidence-based knowledge, not as a personal "belief" (opinion).
- line 361: "It seemed as though debates on the interrelationship among the four main Myriapoda subfamilies could never be settled by phylogenomic analyses alone". Why? Phylogenomic analyses by Szucsich et al. and by the Authors provide a robust phylogeny.
- line 386: "could reflect rapid adaptive changes". Why rapid?

---

## Round 0.2 · accepted · Accept

Dear Dr. Wang and colleagues:

Thanks for revising your manuscript based on the concerns raised by the reviewers. I now believe that your manuscript is suitable for publication. Congratulations! I look forward to seeing this work in print, and I anticipate it being an important resource for groups studying arthropod and myriapod systematics. Thanks again for choosing PeerJ to publish such important work.

Best,

-joe